

# Factors associated with extubation time in coronary artery bypass grafting patients

Abbas Rezaianzadeh[1], Behzad Maghsoudi[2], Hamidreza Tabatabaee[3], Sareh Keshavarzi[3], Zahra Bagheri[4], Javad Sajedianfard[5], Hamid Gerami[2] and Javad Rasouli[3]

[1] Colorectal Research Center, Department of epidemiology, School of Health, Shiraz University of Medical Sciences, Shiraz, Iran
[2] Department of Anesthesia, Shiraz Medical School, Shiraz University of Medical Sciences, Shiraz, Iran
[3] Department of Epidemiology, School of Health, Shiraz University of Medical Sciences, Shiraz Fars, Iran
[4] Department of Biostatistic, Shiraz Medical School, Shiraz University of Medical Sciences, Shiraz, Iran
[5] Department of Physiology, School of Veterinary Medicine, Shiraz University, Shiraz, Iran

Corresponding author
Javad Rasouli, rsljvd@yahoo.com

## ABSTRACT

**Background and Objectives.** Cardiovascular diseases are the leading cause of death worldwide, with coronary artery disease being the most common. With increasing numbers of patients, Coronary Artery Bypass Grafting (CABG) has become the most common operation in the world. Respiratory disorder is one of the most prevalent complications of CABG. Thus, weaning off the mechanical ventilation and extubation are of great clinical importance for these patients. Some post-operative problems also relate to the tracheal tube and mechanical ventilation. Therefore, an increase in this leads to an increase in the number of complications, length of hospital stay, and treatment costs. Since a large number of factors affect the post-operative period, the present study aims to identify the predictors of extubation time in CABG patients using casualty network analysis.

**Method.** This longitudinal study was conducted on 800 over 18 year old patients who had undergone CABG surgery in three treatment centers affiliated to Shiraz University of Medical Sciences. The patients' information, including pre-operative, peri-operative, and post-operative variables, was retrospectively extracted from their medical records. Then, the data was comprehensively analyzed through path analysis using MPLUS-7.1 software.

**Results.** The mean of extubation time was $10.27 + 4.39$ h. Moreover, extubation time was significantly affected by packed cells during the Cardiopulmonary Bypass (CPB), packed cells after CPB, inotrope use on arrival at ICU, mean arterial pressure 1st ICU, packed cells 1st ICU, platelets 1st ICU, Blood Urea Nitrogen 1st ICU, and hematocrit 1st ICU.

**Conclusion.** Considering all of the factors under investigation, some peri-operative and post-operative factors had significant effects. Therefore, considering the post-operative factors is important for designing a treatment plan and evaluating patients' prognosis.

## INTRODUCTION

Cardiovascular diseases are the leading cause of death all around the world. Among these diseases, Coronary Artery Disease (CAD) has been introduced as the first and most common cause of mortality in all age groups (*Archbold & Curzen, 2003*; *Go et al., 2014*; *Yu et al., 2015*). According to the 2010 report by the American Heart Association (AHA), CAD accounted for 1 in every 6 deaths in the US. In addition, 379,555 Americans died as a result of CAD in 2010 (*Go et al., 2014*). Nowadays, CAD is one of the most prevalent diseases resulting in hospitalization in the US. In the United States, more than 13 million individuals suffer from CAD (*Chu et al., 2008*). With an increase in the number of incidents of CAD, Coronary Artery Bypass Grafting (CABG) has become one of the most common operations worldwide, such that almost 500,000 CABG operations are performed in the US every year (*Branca, McGaw & Light, 2001*; *Dorsa et al., 2011*).

Respiratory disorder is one of the most prevalent complications of CABG (*Cohen et al., 2000*; *Faritous et al., 2011*; *Shahbazi & Kazerooni, 2012*; *Totonchi et al., 2014*; *Yende & Wunderink, 2002*). The incident rate of respiratory disorders after CABG has been estimated to be 5–20%, which leads to an annual cost of 2 million dollars (*Christian, Engel & Smith, 2011*). Prolonged Mechanical Ventilation (PMV) after CABG can increase morbidity and mortality rates as well as treatment costs; it may also decrease the quality of life (*Cohen et al., 2000*; *Faritous et al., 2011*; *Shahbazi & Kazerooni, 2012*; *Totonchi et al., 2014*). Advances in science and technology in the field of cardiac surgery have resulted in new techniques for treatment of these disorders. These techniques include new anesthesia methods, open heart surgery without using Cardiopulmonary Bypass (CPB), and less invasive CABG which have somewhat reduced the duration of surgery, extubation time, length of ICU stay, complications, and costs (*Doering, Esmailian & Laks, 2000*).

The present study aims at identifying pre-operative, peri-operative, and post-operative risk factors that relate to extubation time and it aims to determine their effectiveness in the prognosis of patients for increasing the care quality and improving CABG outcomes.

## METHODS

This observational, multicenter study was conducted after gaining approval of the Ethics Committee and Research Vice-Chancellor of Shiraz University of Medical Sciences (proposal No. 93-7247) for the collection of information from patients' medical records. The patients over 18 years old who had undergone open heart surgery in Shahid Faghihi, Al-Zahra, and Kowsar hospitals affiliated to the Shiraz University of Medical Sciences (Shiraz in southern Iran) from April to September 2014 were enrolled into the study. The patients' pre-operative, peri-operative, and post-operative information was retrospectively extracted from their medical records and entered into the study checklist by two trained anesthesia staff of the cardiac operating room (Table 1). In order to ensure the accuracy of the information, the data extracted from 10% of the records was reviewed and matched with the related checklists. The records were re-checked in case of ambiguity. A total of 800 cases overall were entered into the study.

**Table 1  Relevant pre-, perio- and post-operative data collected for all cases.**

| Preoperative data | Perioperative data | Post-operative data |
|---|---|---|
| Sex, Age, Body Mass Index (BMI), Smoking, Addiction, Diabetes Mellitus (DM), Hypertension (HTN), Hyperlipidemia (HLP), Ejection Fraction (EF), Hemoglobin preoperation, chronic obstructive pulmonary diseases (COPD), Hematocrit preoperation, Creatinine preoperation, Blood Urea Nitrogen (BUN) preoperation, prior myocardial infarction (MI) (<30 days) | Anesthesia duration, Operation duration, Mean Arterial Pressure (MAP) pre anesthesia, MAP before cardiopulmonary bypass (CPB), Inotrope use before CPB, Total pump time, Cross clamp time, Urinary output before CPB, Hemofilter volume, Inotrope use during CPB, Inotrope use after CPB, MAP at end of operation, Pack Cell after CPB, Platelet after CPB, activated clotting time/second, Urinary output during CPB, Pack Cell during CPB, Lowest temperature on CPB | Inotrope use arrival to ICU, MAP admission ICU, Urinary output first 1h ICU, Pack Cell 1h ICU, Platelet 1h ICU, Creatinine 1st ICU (arrival to ICU), Blood Urea Nitrogen (BUN) 1st ICU (arrival to ICU), hematocrit 1st ICU (arrival to ICU) |

In this study, the dependent variable was the length of intubation, which was considered to be the period between the patients' arrival at the ICU and their extubation (in hours).

In the recent decades, many attempts have been made towards comprehensive investigation of these variables. One of the most promising methods in this respect is structural equations and multivariate analyses (*Hox, 2010*; *Hoyle, 2014*; *Lleras, 2005*; *Pearl, 2000*).

Investigation of the complex relationships among variables requires utilization of methods, which not only analyze $K$ independent variables and $N$ dependent variables simultaneously, but can also show their mutual effects in a theory-based structure. One such method is a complex mathematical and statistical combination of multivariate regression analysis; i.e., path analysis, which analyzes the variables collected in a complex system (*Hox, 2010*; *Hoyle, 2014*; *Karadag, 2012*; *Lleras, 2005*; *Meehl & Waller, 2002*; *Streiner, 2005*). In the present study, path analysis was done using MPLUS-7.1 software to achieve the objectives and evaluate the intended theoretical model.

During a literature review, it was found that path analysis was performed over 5 stages, namely model formulation, model identification, model estimation, model evaluation, and model modification. Moreover, Root Mean Square Error of Approximation (RMSEA), Tucker-Lewis Index (TLI), Comparative Fit Index (CFI), and Square Residual Standardized Root Mean (SRMR) were used to assess the appropriateness of the designed model (*Wang & Wang, 2012*). Pearson and Spearman correlation coefficients were used for univariate analyses and the means were compared by an independent sample $t$-test. $P < 0.05$ was considered as statistically significant in all tests.

In order to carry out path analysis, first a theoretical model should be designed based on the previous findings and researchers' assumptions to provide the basis for analyses. The theoretical model based on pre-operative, peri-operative, and post-operative stages has been provided with deep literature review, where the effects of different variables on the dependent variable (extubation time) can be determined.

## RESULTS

This longitudinal study was conducted on 800 patients who had undergone CABG surgery. The results showed that the patients' ages ranged from 20 to 89 years old with a mean age of

**Table 2 Baseline quantitative characteristics of the patients and univariate analysis result with extubation time.**

| Risk factors | Mean ± SD | Pearson correlation coefficient | p-value |
|---|---|---|---|
| Age[*] | 59.26 ± 11.60 | 0.25 | <0.001 |
| Body Mass Index[*] | 25.75 ± 4.18 | −0.17 | <0.001 |
| Ejection fraction[*] | 49.19 ± 10.51 | −0.09 | 0.01 |
| Hemoglobin preoperation | 11.81 ± 1.78 | −0.07 | 0.07 |
| Hematocrit preoperation[*] | 35.04 ± 5.37 | −0.08 | 0.03 |
| Creatinine preoperation[*] | 1.08 ± 0.35 | 0.12 | 0.00 |
| BUN Preoperation[*] | 17.54 ± 6.29 | 0.20 | <0.001 |
| Anesthesia duration[*] | 4.49 ± 0.73 | 0.08 | 0.02 |
| Operation duration | 3.08 ± 0.70 | 0.06 | 0.12 |
| Mean arterial pressure pre anesthesia[*] | 99.93 ± 14.28 | −0.09 | 0.01 |
| MAP before CPB | 71.25 ± 12.21 | −0.01 | 0.82 |
| Total pump time[*] | 70.38 ± 20.71 | 0.10 | 0.01 |
| Cross clamp time | 40.12 ± 13.55 | 0.03 | 0.39 |
| Urinary output before CPB | 240.54 ± 251.94 | −0.01 | 0.88 |
| Hemofilter volume | 1588.74 ± 776.33 | 0.02 | 0.57 |
| MAP at end of operation | 75.93 ± 9.60 | −0.06 | 0.07 |
| Pack Cell after CPB[*] | 0.48 ± 0.60 | 0.11 | 0.00 |
| Platelet after CPB | 0.06 ± 0.51 | 0.05 | 0.19 |
| Urinary output during CPB | 532.18 ± 394.18 | −0.06 | 0.12 |
| Pack Cell during CPB[*] | 0.88 ± 0.85 | 0.14 | <0.001 |
| Lowest temperature on CPB[*] | 33.04 ± 1.16 | −0.08 | 0.03 |
| MAP admission ICU[*] | 79.48 ± 14.77 | −0.18 | <0.001 |
| MAP on 6h ICU[*] | 79.10 ± 10.67 | −0.15 | <0.001 |
| Urinary output first 6h ICU[*] | 1403.67 ± 614.89 | −0.16 | <0.001 |
| Platelet 1h ICU[*] | 0.25 ± 0.99 | 0.12 | 0.00 |
| Creatinine 1st ICU (arrival to ICU)[*] | 1.00 ± 0.31 | 0.17 | <0.001 |
| BUN 1st ICU (arrival to ICU)[*] | 16.15 ± 5.91 | 0.23 | <0.001 |
| Hct 1st ICU (arrival to ICU)[*] | 31.85 ± 4.37 | −0.15 | <0.001 |
| Extubation hours after arrival in ICU | 10.27 ± 4.39 | – | – |

**Notes.**

[*] Statistically significant (p-value < 0.05).

59.26 ± 11.60 years. Most of the patients 483 (60.4%) were male. The mean of extubation time was 10.27 ± 4.39 h. The patients' basic characteristics have been presented in Table 2. In addition, the results of the univariate analysis of qualitative and quantitative risk factors of the study have been shown in Tables 3 and 2.

As Table 2 depicts, some factors correlated significantly with extubation time. The results of univariate analysis also indicated the great importance of pre-operative and post-operative risk factors.

The results of the independent sample t-test (Table 3) revealed that the mean of extubation time was higher in females and in patients with diabetes, hypertension, and Chronic Obstructive Pulmonary Disease (COPD), but the difference was not statistically

**Table 3** Baseline qualitative characteristics of the patients and univariate analysis result with extubation time.

| Risk factors | Mean ± SD | p-value |
|---|---|---|
| Sex (Male/Female) | 10.1 ± 4.18/10.52 ± 4.68 | 0.186 |
| Smoking (yes/no) | 9.87 ± 4.32/10.42 ± 4.40 | 0.116 |
| Addiction (yes/no) | 9.71 ± 4.37/10.37 ± 4.38 | 0.124 |
| Diabetes Mellitus (yes/no) | 10.31 ± 4.62/10.24 ± 4.27 | 0.824 |
| Hypertension (yes/no) | 10.48 ± 4.33/9.91 ± 4.47 | 0.076 |
| Hyperlipidemia (yes/no) | 10.22 ± 4.69/10.31 ± 4.08 | 0.762 |
| Chronic obstructive pulmonary diseases (yes/no) | 10.43 ± 5.03/10.26 ± 4.36 | 0.831 |
| Any arrhythmia (yes/no) | 10.49 ± 4.22/10.24 ± 4.40 | 0.629 |
| MI <30 days (yes/no) | 9.65 ± 4.43/10.35 ± 4.37 | 0.142 |
| Inotrope use before CPB (yes/no)[*] | 13.15 ± 6.33/10.12 ± 4.21 | <0.001 |
| Inotrope use during CPB (yes/no)[*] | 10.89 ± 5.14/9.89 ± 3.83 | 0.002 |
| Inotrope use after CPB (yes/no)[*] | 10.61 ± 4.71/9.68 ± 3.71 | 0.003 |
| Inotrope use arrival to ICU (yes/no)[*] | 10.72 ± 4.67/9.46 ± 3.51 | <0.001 |
| Inotrope use on 6h ICU (yes/no)[*] | 11.83 ± 5.12/ 9.32 ± 3.5 | <0.001 |

**Notes.**

[*] Statistically significant ($p$-value < 0.05).

significant ($P > 0.05$). Conversely, the mean of extubation time was significantly higher in the patients who had received inotrope compared to those who had not ($P < 0.001$). In addition, the mean age of the patients who had received inotrope was significantly higher in comparison to those who had not ($P < 0.037$). Considering multiple relations and the probability of the impact of confounding factors, the interpretation of univariate analysis results should be done with due caution. In order to control the effects of the confounding factors and assess the multiple associations, path analysis was used and the results were presented in Table 4. In path analysis, direct, indirect, and total effects can be evaluated and confounding effects can be controlled. Therefore, the results can be applied with greater certainty. The variables with one significant effect (direct, indirect, or total) have been shown in Table 4. It should be noted that total effects are more important on the basis of decision-making. Furthermore, values related to effect size of the study's variables are in fact the standardized coefficient, which is used for similarity and comparability of the measurement units of all the variables. Interpretation of this coefficient is the same as that of the regression coefficient and its value varies between −1 and +1.

According to the results of path analysis, presented in Table 4, packed cells during CPB and packed cells after CPB (among peri-operative variables) and inotrope use on 1h ICU (arrival to ICU), mean arterial pressure on 1h ICU, pack cell 1h ICU, platelet 1h ICU, Blood Urea Nitrogen (BUN) 1h ICU, and hematocrit 1st ICU (among post-operative variables) were effective on extubation time ($P < 0.05$). Yet, some of the risk factors had significant direct or indirect effects on extubation time, which were modified in computation of the total effects.

The final model of the relationships among the factors affecting the dependent variable (extubation time) and their effect paths have been presented in Fig. 1. It should be

**Table 4 Standardized direct, indirect and total risk factors effects on extubation time.**

| Risk factor | Direct effect | p-value | Indirect effect | p-value | Total effect | p-value |
|---|---|---|---|---|---|---|
| Sex | 0.014 | 0.779 | 0.056 | 0.011[*] | 0.070 | 0.123 |
| Diabetes Mellitus | −0.094 | 0.017[*] | 0.019 | 0.037[*] | −0.076 | 0.058 |
| Pack Cell during CPB | 0.191 | <0.001[*] | −0.017 | 0.105[*] | 0.174 | 0.001[*] |
| Hemoglobin preoperation | 0.126 | 0.388 | −0.099 | 0.020[*] | 0.027 | 0.853 |
| Pack Cell after CPB | 0.102 | 0.008[*] | −0.006 | 0.254 | 0.096 | 0.013[*] |
| Hypertension | 0.035 | 0.403 | 0.021 | 0.027[*] | 0.056 | 0.181 |
| Bun 1st ICU | 0.128 | 0.019[*] | 0.02 | 0.325 | 0.147 | 0.004[*] |
| HCT 1st ICU | −0.091 | 0.038[*] | – | – | −0.091 | 0.038[*] |
| Inotrope use arrival to ICU | 0.129 | 0.039[*] | – | – | 0.129 | 0.039[*] |
| MAP 6h ICU | −0.123 | 0.002[*] | – | – | −0.123 | 0.002[*] |
| Platelet 6h ICU | 0.119 | 0.003[*] | – | – | 0.119 | 0.003[*] |
| Pack Cell 6h ICU | 0.115 | 0.007[*] | – | – | 0.115 | 0.007[*] |

**Notes.**

[*] Statistically significant (p-value < 0.05).

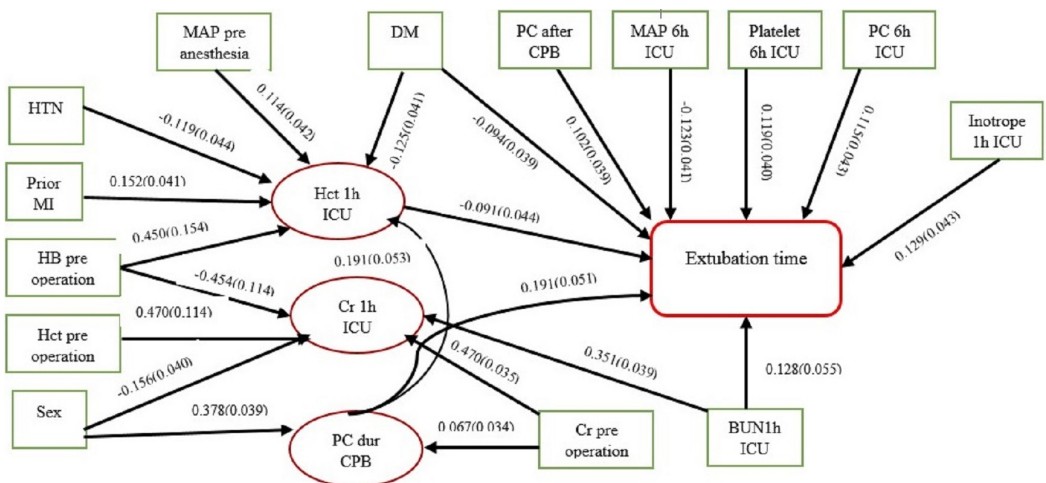

**Figure 1 Causal network diagram of influenced factors of extubation time.** (RMSEA = 0.036, CFI = 0.910, TLI = 0.901, SRMR = 0.016). * The arrow indicator values are standardized direct effects with standard deviation.

mentioned that the path coefficients representing the direct effects, and standard deviation of each variable in each corresponding path, have been shown in this model. Mediator variables have also been determined.

Based on the software output, RMSEA = 0.036, CI 0.90 [0.021–0.046], CFI = 0.910, TLI = 0.901, and SRMR = 0.016. Considering the proposed values for decision-making (TLI/CFI ∼ 1 and SRMR/RMSEA < 0.05) (*Wang & Wang, 2012*), the designed model had acceptable appropriateness.

According to the number of variables of the study, to ensure enough power of the study, using the R software for power analysis, the results showed that the power of the study was 0.81, indicating its adequacy are the recommended amounts.

## DISCUSSION

In the present study, none of the pre-operative factors had significant impacts on extubation time. Among these variables, sex and previous history of diabetes mellitus, hypertension, and hemoglobin pre-operation had significant direct or indirect effects on the variable under study. However, considering various effects with different paths and directions, their total effects were not statistically significant (Table 4). In the study by *Suematsu et al. (2000)*, the results of univariate analysis indicated that none of the preoperative factors (age, sex, Body Mass Index (BMI), smoking, hypertension, hyperlipidemia, diabetes, COPD, renal disease, liver disease, and EF) had significant effects. Age was only found to be significantly effective in multivariate analysis. Similarly, *Christian, Engel & Smith (2011)* reported that COPD, diabetes, hypertension, sex, and BMI had significant effects in univariate analysis of the pre-operative variables, but only sex was significantly effective in multivariate analysis.

The findings of univariate analysis in the current study showed that, among the peri-operative factors, anesthesia duration, pre-anesthesia mean arterial pressure, total pump time, inotrope use before CPB, inotrope use during CPB, inotrope use after CPB, packed cells after CPB, packed cells during CPB, and lowest temperature on CPB had significant effects. However, only packed cells during CPB and packed cells after CPB had significant effects in the final model (Table 4).

In the research *Christian, Engel & Smith (2011)*, the results of univariate analysis of peri-operative factors demonstrated that anesthesia time, operation time, lowest temperature, and transfusion were significantly effective. In multivariate analysis, however, only anesthesia time had a significant effect.

Considering the post-operative factors, the results of univariate analysis demonstrated the significant effects of inotrope use on arrival to ICU, inotrope use on 1h ICU, mean arterial pressure on ICU admission, urinary output 1h ICU, packed cells 1h ICU, platelets 1h ICU, creatinine 1st ICU, BUN 1st ICU, and hematocrit 1st ICU. Only inotrope use on 1h ICU, mean arterial pressure on 1h ICU, packed cells 1h ICU, platelets 1h ICU, BUN 1st ICU, and hematocrit 1st ICU had significant effects in path analysis (Table 4).

Other studies have also revealed the impacts of transfusion and inotrope use on extubation time (*Christian, Engel & Smith, 2011*; *Faritous et al., 2011*; *Scott, Seifert & Grimson, 2008*; *Shahbazi & Kazerooni, 2012*; *Suematsu et al., 2000*; *Totonchi et al., 2014*). Up to now, there has been much debate on the risks of packed red blood cell transfusion. For instance, a review study by David and Gerber (*Gerber, 2012*) demonstrated the high prevalence of packed red blood cell transfusion, post-operative mechanical ventilation, and its related respiratory disorders in CABG patients, which is in line with the results of the present study. Similarly, *Vamvakas & Carven (2002)* conducted a study on 416 CABG patients to assess the effects of packed red blood cell transfusion and platelet, plasma, and total fluid volumes on post-operative mechanical ventilation. The study's results revealed that only packed red blood cell transfusion was significantly associated with post-operative mechanical ventilation.

Evidence has indicated that some factors, such as age, sex, respiratory status, and BMI, impact on extubation time in patients undergoing open heart surgery (*Chu et al., 2008*; *Shahbazi & Kazerooni, 2012*). Yet, contradictory results have been obtained in this regard from different studies. For instance, some studies have shown the role of age and sex, but some others have not (*Azarfarin et al., 2014*; *Forouzannia et al., 2011*; *Ghotkar et al., 2006*; *Hein et al., 2006*). The difference in the results could be attributed to differences in methodologies, statistical methods, and sample sizes in various studies. Moreover, the physiological factors under investigation have multiple relations and may cause both positive and negative effects. Therefore, not accurately controlling the confounding variables could deflect the results. The researchers of the present study considered a large number of pre-operative, peri-operative, and post-operative factors, which were thought to have probable effect, and considered all the associations as a casualty network, and took all the mutual effects into account to provide a proper interpretation of the relationships. In complex relations, each variable can play the role of dependent and independent variables at the same time, and path analysis has the capability of such presuppositions (*Lleras, 2005*; *Meehl & Waller, 2002*; *Wang & Wang, 2012*). Thus, in addition to having effect through different paths, a factor can have both positive and negative effects in a network. In this case, the outcome of these effects determines the factor's final impact on the final outcome.

The findings of our study showed that utilization of blood products and inotrope could increase extubation time, while the measures increasing the patients' mean arterial pressure and hematocrit in ICU, could decrease this time in the CABG patients.

Some researchers have referred to the impact of differences between the surgeon and anesthesiologist on the operation outcome. However, this was not taken into account in this study due to some limitations. Hence, further studies with larger sample sizes are recommended to be conducted in other centers using similar and even more advanced analytical methods in order to gain a deeper understanding of extubation time prognostic factors. In addition, future studies are suggested to be performed on the complications of delayed extubation, its effective factors, and contribution of each factor to different stages of the process to prevent unpleasant outcomes.

## ACKNOWLEDGEMENTS

This article was extracted from a PhD dissertation in Epidemiology approved by Shiraz University of Medical Sciences. Hereby, the authors would like to thank the Research Vice-Chancellor of the University and personnel and authorities of the study hospitals for their cooperation in data collection.

### Funding

This work was not funded by any external sources.

### Competing Interests

The authors declare there are no competing interests.

## Author Contributions

- Abbas Rezaianzadeh and Behzad Maghsoudi conceived and designed the experiments, performed the experiments, contributed reagents/materials/analysis tools, wrote the paper, reviewed drafts of the paper.
- Hamidreza Tabatabaee conceived and designed the experiments, contributed reagents/materials/analysis tools, wrote the paper, prepared figures and/or tables, reviewed drafts of the paper.
- Sareh Keshavarzi analyzed the data, contributed reagents/materials/analysis tools, prepared figures and/or tables.
- Zahra Bagheri analyzed the data, contributed reagents/materials/analysis tools.
- Javad Sajedianfard performed the experiments, wrote the paper, reviewed drafts of the paper, data collection.
- Hamid Gerami data collection.
- Javad Rasouli conceived and designed the experiments, performed the experiments, analyzed the data, contributed reagents/materials/analysis tools, wrote the paper, prepared figures and/or tables, reviewed drafts of the paper.

## Human Ethics

The following information was supplied relating to ethical approvals (i.e., approving body and any reference numbers):

This work was approved by the Ethics Committee and Research Vice-chancellor of Shiraz University of Medical Sciences. The PhD dissertation in Epidemiology that this work draws from was approved by Shiraz University of Medical Sciences (proposal No. 93-7247).

## Supplemental Information

Supplemental information for this article can be found online at http://dx.doi.org/10.7717/peerj.1414#supplemental-information.

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
