# Peer review of "Factors associated with extubation time in coronary artery bypass grafting patients"

_PeerJ, doi:10.7717/peerj.1414_

## Round 0.1 · original submission · Minor Revisions

As indicated by the two reviewers, an extensive analysis of the factors effecting the extubation time in CABG patients using causality network analysis is valuable information. During you resubmission of the manuscript, please address the comments related to clarity in discussion and conclusions (concern raised by reviewer 1). Also, please address minor concerns raised by reviewer 1 and 2.
Thanks

Reviewer 1 ·

Basic reporting

No Comments

Experimental design

Experiments were well designed

Validity of the findings

Well represented!
Elaborate on the conclusion part.

Additional comments

The present study aims to identify the predictors of extubation time in CABG patients using casuality network analysis.As identifying crucial factors will aid in designing a treatment plan for better evaluating patient prognosis. Authors have performed an extensive analysis in 800 patients undergone CABG.

I have only few concerns

-Reframe the conclusion part in the abstract,as it is unclear. please elaborate
-Authors should provide references for the text mentioned in the lines 195 &196
-Include conclusion after the discussion part.
-Authors may include a better representation of the Fig-1,as it is difficult to extrapolate the message from the present figure.
-Authors should clarify about the lines 218 & 219. As it is unclear
- Few sentences were repeated in results as well as in discussion the part. Authors might want to reframe the same

Reviewer 2 ·

Basic reporting

The introduction is insufficient to give enough background about the study. As per the journal requirement, raw data should be provided with the manuscript. Hence, Please provide the raw data.

While the clinical data is collected from hundreds of patients and various statistical methods are used to analyse the data, more importance is given throughout the paper to the statistical methods used and the focus on the actual clinical issue of respiratory problems associated with coronary artery bypass grafting is lacking in this paper.

The result section can be expended and more clearly written to explain the findings of the statistical analysis and key findings.

The discussion section mostly restate the results and it can be written more clearly to explain how the findings of the study are related to the respiratory problems in coronary artery bypass grafting.

Other minor comments are,

Line 32. Period should be used instead of colon after Background and Objectives

Line 42. Period should be used instead of colon after Method.

Line 47. Period should be used instead of colon after results. Plus or minus sign should be used for between the mean and standard deviation of the extubation time.

Line 51. Period should be used instead of colon after conclusion.

Line 116. Plus or minus sign should be used for between the mean and standard deviation of the age.

Line 117. Plus or minus sign should be used for between the mean and standard deviation of the extubation time.

Experimental design

Data are collected from 800 patients and several statistical methods are used to analyse the data. However, more importance is given to statistical method rather than the actual clinical problem of respiratory issue.

Validity of the findings

no comments

Additional comments

While the clinical data is obtained from about 800 patients and analyzed using different statistical methods, overall the manuscript is poorly written and lacks to discuss about the major clinical issues.

---

## Round 0.2 · accepted · Accept

After careful consideration of the original reviews and the present comments from the reviewers, it seems that the authors have satisfactorily addressed all the concerns raised by the reviewers. I am looking forward to the publication of the manuscript.

Reviewer 1 ·

Basic reporting

Good

Experimental design

Good

Validity of the findings

Good

Additional comments

Authors have made necessary suggested corrections and looks to good to me now

Reviewer 2 ·

Basic reporting

No comments and the authors implemented the suggestions given during the previous review process.

Experimental design

No comments

Validity of the findings

Clinically relevant information is concluded in this study.